# Predictors of COVID-19 vaccine uptake among persons aged 18 years and above in Ga North Municipality, Ghana using the Health Belief Model: A community-based cross-sectional study

Imoro Nasiratu[1], Lilian Belole Pencille[2], Nelisiwe Khuzwayo[3], Richard Gyan Aboagye[4], Elvis Enowbeyang Tarkang[1,2,3]*

1 Department of Population and Behavioural Sciences, School of Public Health, University of Health and Allied Sciences, Ho, Ghana, 2 HIV/AIDS Prevention Research Network Cameroon, Kumba, Cameroon, 3 Discipline of Public Health Medicine, School of Nursing and Public Health, University of KwaZulu-Natal, Durban, South Africa, 4 School of Public Health, University of Health and Allied Sciences, Ho, Ghana

* ebeyang1@yahoo.com

## Abstract

### Background

Although the coronavirus disease 2019 (COVID-19) vaccination rollout has been accepted by the population of the Ga North Municipality, a substantial proportion has developed hesitancy to COVID-19 vaccination uptake. This study determined the predictors of COVID-19 vaccine uptake among persons aged 18 years and above in the Ga North Municipality using the Health Belief Model.

### Methods

The study used a cross-sectional study design. Structured questionnaires were used to collect data from a multistage sample of 388 respondents. Multivariable binary logistic regression was used to determine the predictors of COVID-19 vaccination uptake at the level of 0.05 and 95% confidence interval.

### Results

Vaccination uptake was 72.2%. The odds of COVID-19 vaccination uptake were higher among men than women [AOR = 2.02, 95% CI: 1.13–3.20] and among singles than the married [AOR = 1.90, 95% CI: 1.07–3.36], but lower among Muslims than Christians [AOR = 0.33, 95%CI: 0.18–0.60]. Perceived susceptibility [AOR = 2.43, 95% CI: 1.36–4.35], perceived barriers [AOR = 0.54, 95%CI: 0.31–0.95], cues to action [AOR = 2.23, 95% CI: 1.19–4.21] and self-efficacy [AOR = 3.23 95% CI: 1.82–5.71] were the significant predictors of COVID-19 vaccination uptake.

**Data Availability Statement:** All relevant data are within the manuscript and its Supporting Information files.

**Funding:** The authors received no specific funding for this work.

**Competing interests:** The authors have declared that no competing interests exist.

## Conclusion

The uptake of the COVID-19 vaccine in GA North Municipality is high. Health promotion interventions should focus on increasing perceived susceptibility to COVID-19, minimising barriers to COVID-19 vaccine uptake, and promoting cues and self-confidence for COVID-19 vaccine uptake. It should also target women, the married, and Muslims.

## Background

The Coronavirus disease 2019 (COVID-19) has had a negative toll on the health of populations globally. Since its identification in Wuhan, China and the subsequent declaration as a public health emergency of international concern (PHEIC), no country has been spared [1]. Countries have adopted and implemented non-pharmacological measures recommended by the World Health Organization (WHO), including border closure, hand washing with soap and clean running water, physical distancing, mandatory wearing of face masks, debarment of public gatherings, and intensification of surveillance system [2, 3]. However, as of 27 June 2021, global case prevalence and mortality were more than 180 million and 3.9 million respectively [4].

Notwithstanding the interventions instituted, Ghana has recorded 95,642 cases and 795 deaths as of 27[th] June 2021 [5]. All facets of the country have been negatively affected, from the social to the economic life of its population. These negative impacts of the pandemic have increased concerns about global efforts to attain Sustainable Development Goal 3 (SDG 3). Consequently, studies were conducted to ascertain the natural conferment of immunity upon infection with SARS-CoV-2, and it was evidenced that more than 90% of infected persons developed immunity. However, the duration and possibility of preventing re-infection remained uncertain for individuals with previous infection with SARS-CoV-2 [6]. Therefore, the need to reduce the basic reproductive rate ($R_0$) to less than 1 was tantamount to immunisation, to achieve herd immunity. Immunisation has helped save millions of lives. It has helped control, eliminate, and eradicate diseases with high morbidity and mortality rates globally. In addition to these direct benefits of immunisation, it has impacted positively on the socioeconomic status of families [7].

In Africa, immunisation coverage has improved significantly [8]. The introduction of COVID-19 vaccination has generally decreased COVID-19 cases and mortality globally except in a few countries where high community transmission continues to exist [4]. Despite the proven efficacy of COVID-19 vaccines, uptake of the COVID-19 vaccination has not been encouraging particularly in low-and-middle-income countries (LMICs) [9]. Ghana has procured more than one million vaccine doses with only 1.2% population coverage as of 23[rd] June 2021 as against the set target of vaccinating at least 20 million of its population [10]. This poor coverage has been associated with a combination of vaccine unavailability and poor uptake of COVID-19 vaccines, influenced by an array of demographic, sociocultural, and health-related factors [11, 12]. Poor uptake of the COVID-19 vaccine has been a major canker that has led to the inability of countries to achieve recommended immunisation coverage rates and has been described by WHO as one of the top ten threats to public health [13].

Greater Accra is the major epicentre of COVID-19 in Ghana. It has recorded more than half of the country's total case prevalence [5]. Ga North is one of the hardest-hit Municipalities within the Greater Accra Region of Ghana. The Municipality recorded 541 cases prevalence with a 6.3% case fatality rate (CFR) as of 27[th] June 2021 [14]. The first phase of the vaccination

roll-out programme in the Municipality saw less than 75% uptake among the population [14]. It is, therefore, crucial to identify and understand the predictors of COVID-19 vaccine uptake among the population to promote high vaccination uptake and achieve herd immunity in the second phase of the vaccination programme. This study aims to determine the predictors of COVID-19 vaccine uptake using the Health Belief Model (HBM).

In the phase of educational campaigns, lockdown, and procurement and distribution of personal protective equipment at various institutions, community transmission continues to exist in the Municipality. Factors assumed to contribute to poor uptake of COVID-19 vaccination include sociodemographic factors and perceptions about COVID-19 and COVID-19 vaccines [15]. Low COVID-19 vaccine uptake in the community will lead to the accumulation of vulnerable populations and increase the risk of entrenching community transmission of COVID-19. This phenomenon defeats the Municipality's efforts to achieve herd immunity among its population to decrease morbidity and mortality from COVID-19. It is against this background that this study was conducted in the Municipality to identify and understand the predictors of COVID-19 vaccination uptake using the HBM as the conceptual framework.

## Conceptual framework

The Health Belief Model is one of the dominant theories used to understand factors that influence decision-making by assessing the motivators and barriers underlying health-seeking behaviour [16, 17]. The model has six constructs, namely perceived susceptibility to a health outcome (the subjective perception of the risk of contracting COVID-19), perceived severity of the health outcome (one's feelings about the clinical, medical and social consequences of contracting COVID-19), perceived benefits of the recommended health-related action (COVID-19 vaccine uptake), perceived barriers or cost of taking the recommended health action (tangible and psychological costs associated with COVID-19 vaccine uptake), cues to to take the recommended health action (information and media campaigns encouraging COVID-19 vaccine uptake), and self-efficacy or perceived competence in taking the recommended health action (confidence in ability to uptake COVID-19 vaccine). In the context of the current study, the predictors of COVID-19 vaccine uptake would be the constructs of the HBM (Fig 1) [17].

Socio-demographic variables such as health status, vaccination history, COVID-19 experience, and knowledge of COVID-19 and its vaccine may influence individuals' perceived susceptibility to and severity of SARS-CoV-2. Awareness of the vulnerability and complications that characterise SARS-CoV-2 infection may influence the decision to uptake the COVID-19 vaccine. These variables may also influence individuals' perceptions of the benefits of and barriers to vaccine uptake. Cost-benefit analysis and attainment of a positive perception of the ability of vaccination to prevent diseases with minimal cost (physical, material, or psychological) may influence the decision to uptake the COVID-19 vaccine. Vaccination history and acceptance of vaccination by relatives may influence a person's decision to uptake the COVID-19 vaccine.

The HBM has been used as a theoretical framework to investigate COVID-19 vaccine acceptance and intention to accept in China [18], Malaysia [19], Hongkong [20] and Ghana [21]. However, due to differences in demographic characteristics, perceptions and environmental factors, and also to the fact that the studies that used the HBM, investigated the willingness to accept the COVID-19 vaccine if provided, the current study determined the predictors of COVID-19 vaccine uptake among persons aged 18 years and above in Ga North Municipality, Ghana using the HBM as the theoretical framework.

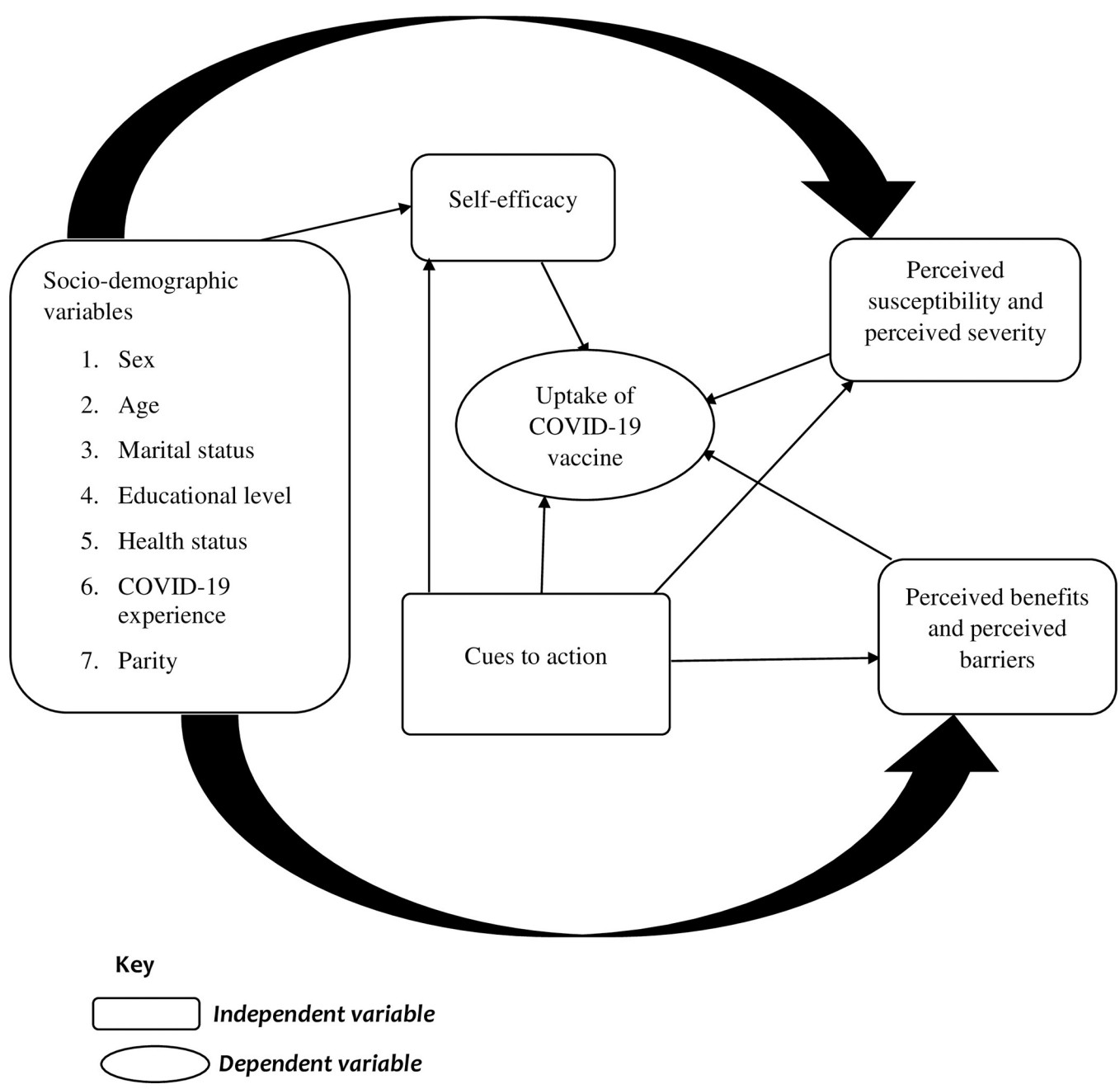

**Fig 1. Conceptual framework, adapted from Abraham & Sheeran (2016) [17].**

## Methods

### Study design

A community-based quantitative analytical cross-sectional study design was used for the study. This design was used to determine the predictors of COVID-19 vaccination uptake among the defined population of the study. However, the design was not able to measure future or past factors that influence COVID-19 vaccination uptake.

## Description of the study site

The Ga North Municipality was carved out of the Ga West Municipality, which was created in 2018 in pursuance of the government's decentralisation and local government reform policy. The Municipality has a population of 135,194 with males constituting a slight majority (50.3%) and a population growth of 4.4%. It is subdivided into 4 health catchment areas, referred to as Health sub-municipalities namely Pokuase, Ofankor, Trobu, and Amanfrom. It is bounded to the north by Ga West Municipality, Accra Metropolitan to the south, Ga Central/Ablekuma North Municipality to the west, and Ga East Municipality to the east. Most of the people living in the Municipality are formal workers who are mostly teachers, businessmen and women and a few artisans.

The Ga North Municipality has more than one thousand churches. They also have various religious faiths. Some of the churches admit clients with various ailments for prayer and spiritual healing thus, indicating a high belief system in the Municipality.

In terms of healthcare, the Municipality is made up of four sub-municipalities with various health staff, and work in close collaboration with private facilities in providing essential healthcare services through the following facilities: one Municipal Hospital, four health centres, four community clinics, fifteen urban Community-Based Health Planning Services (CHPS) Zones, and twenty-two private facilities. COVID-19 vaccination rollout is conducted in all the sub-municipalities. Ofankor has two vaccination sites, Trobu has four, and Pokuase and Amanfrom have three each. However, the Municipality has 12 identified hard-to-reach areas [22].

## Study population

The study targeted persons 18 years and above living in the Ga North Municipality of Ghana.

## Eligibility criteria

All persons 18 years and above and who had lived in the Municipality for at least six months, and were present at the time of the study, were included in the study after giving their written informed consent.

## Sample size determination

A minimum sample size was determined using Cochran's formula [23].

$n = Z^2 \times p \times (1-p)/e^2$

where **n** = minimum sample size, **Z** = Z score at 95% confidence interval (1.96), **p** = prevalence of COVID-19 vaccine uptake of 40% in a study conducted in Kenya [24], and **e** = 5% margin of error.

$n = 1.96^2 \times 0.4 \times (1–0.4)/0.05^2 = 369$

Adding a non-response rate of 5% [25] = (0.05 x 369) + 369, gave a sample size of **388**. Therefore, the study included 388 persons who met the inclusion criteria.

## Sampling procedure

A multistage sampling technique was used to select the respondents of the study. First, the Municipality was divided into four clusters (sub-municipalities). A proportional sample was estimated for each sub-municipality based on the population of the entire Municipality and the sub-municipalities as follows:

Pokuase sub-Municipality = 32447/135194 x 388 = 93
Amanfrom sub-Municipality = 21631/135194 x 388 = 62
Ofankor sub-Municipality = 51374/135194 x 388 = 147

Trobu sub-Municipality = 29743/135194 x 388 = 86

The second stage involved the selection of communities. Simple random sampling was used to select two communities from each sub-municipality. The selection process involved writing the names of all communities on separate pieces of paper. The papers were subsequently folded, placed in a bowl, and shuffled. Trained research assistants were made to select two papers, one after the other with replacements. If a community was selected for the second time, it was placed back in the bowl, reshuffled, and reselected. The required sample size for each sub-municipality was proportionally divided among the two selected communities. The third stage involved the selection of the households. A list of households was obtained from the Municipal Assembly to serve as a sampling frame for each selected community. Systematic random sampling was used to select households from each community. It involved estimating the sampling interval as N/n, where N is the number of households in the community and n is the required sample size for the community. The fourth stage was the selection of the participants for the study. At the household level, simple random was used to select the participants. At each household level where more than three eligible participants exist, simple random sampling was used to select three participants for the study. Selected households without eligible participants were skipped to the next household.

## Data collection tool and procedure

Data were collected using a structured questionnaire adapted from previous studies [16, 26, 27]. The questionnaire was in three sections: sociodemographic factors, COVID-19 vaccine uptake and the HBM constructs. The internal consistency of the items used to measure the constructs of HBM was ensured using the Cronbach's alpha coefficient. The results from Table 1 show that the overall alpha coefficient for the entire HBM was 0.8554, ranging from 0.6759 (for perceived severity) to 0.8626 (for cues to action), which is adequate [28]. The coefficients were calculated using the best combination of items under each construct of the HBM, excluding the response options. Cronbach's alpha does not provide a very good estimate when the items making up the measurement scale are small in number. The more items to measure a construct the more accurate the coefficient [29]. Therefore, the relatively low alpha estimated for perceived severity (0.6759), may be due to the small number of items measuring this construct (3 items).

Data collection was done at the comfort of the respondents' household in October-November, 2022. Pretesting of the questionnaire was done in a conveniently selected community that was excluded from the main study. All participants who could read and write were given the questionnaire to respond to. However, participants who could not read and write were assisted by a research assistant, in the presence of an impartial witness selected by the participants.

**Table 1. Scale reliability coefficient of HBM constructs.**

| Construct | Internal consistency |
|---|---|
| Perceived susceptibility | 0.8035 |
| Perceived severity | 0.6759 |
| Perceived benefits | 0.7499 |
| Perceived barriers | 0.7649 |
| Cues to action | 0.8626 |
| Self-efficacy | 0.8438 |
| **Overall** | **0.8554** |

## Measures

**Dependent variable.** This was measured by asking the respondents whether they had received the COVID-19 vaccine or not with 'Yes' or 'No' as the response options.

**Independent variables.** The independent variables were the sociodemographic variables and the constructs of the HBM. Regarding the constructs of the HBM, perceived susceptibility was measured using four items: 'I can contract COVID-19', 'Increased chances of COVID-19 if unvaccinated', 'Increased chances of infecting family members if unvaccinated', and 'Fear of getting infected with COVID-19'. Perceived severity was measured using three items: 'Low chances of recovering from COVID-19 if infected', 'Develop complications if infected with COVID-19', and 'Likely to die if infected with COVID-19'. Perceived benefit was measured using two items: 'Prevention against COVID-19 if vaccinated' and 'Protection from developing complications if vaccinated'. The perceived barrier was measured using six items: 'Far distance to vaccination site is a barrier to COVID-19 vaccination', 'Side effects of vaccines', 'Vaccines are ineffective', 'I have busy schedules', 'Mistrust of vaccines', and 'COVID-19 vaccines cause complications'. Cues to action was measured using five items: 'I have enough information on COVID-19 and its vaccines', 'Acceptance of COVID-19 vaccination when given adequate information on COVID-19 and its vaccines' 'Acceptance of COVID-19 vaccination if opinion leaders express benefits of vaccines' 'Acceptance of vaccination if family and friends express support for the benefits of vaccines' and 'Acceptance of vaccination if mandatory'. Self-efficacy was measured using three items: 'Willingness to vaccinate if COVID-19 vaccine is made available', 'Confidence to vaccinate despite the side effects of vaccines', and 'Confidence to vaccinate despite the reluctance of opinion leaders, family and friends to vaccinate'. All the items were measured on a 4-point Likert scale (Strongly agree, agree, disagree and strongly disagree). During the statistical analysis, 'strongly agree' and 'agree' were considered as an agreement to an item while 'strongly disagree' and 'disagree' were considered as a disagreement to an item.

## Data analysis

Data were analysed using Stata version 13. Percentage was used to summarise the results of the COVID-19 vaccine uptake. Also, percentages were used to present the results of the COVID-19 vaccine uptake per construct of the HBM. Chi-square analysis was performed to test the association between categorical variables, while binary logistic regression was used to determine the strength of the association between the dependent variable (COVID-19 vaccine uptake) and the independent variables (sociodemographic variables & the constructs of the HBM) at a significance level of 0.05 and 95% confidence interval. Binary logistic regression predicts the probability that an observation falls into one of two categories of a dichotomous dependent variable (COVID-19 vaccine uptake) based on one or more independent variables that can be either continuous or categorical (sociodemographic variables & the constructs of the HBM).

For binary logistic regression to be performed, the following assumptions must be met:

- The dependent variable should be measured on a dichotomous scale

- One or more independent variables which can be either continuous (interval or ratio variable) or categorical (ordinal or nominal)

- The independence of observations and the dependent variable should have mutually exclusive and exhaustive categories.

The procedure gives rise to estimates of the odds of a certain event occurring (COVID-19 vaccine uptake), given a set of explanatory variables (the constructs of the HBM). All these

assumptions were fulfilled in the current study, thus justifying the use of binary logistic regression.

All the statistically significant independent variables (the constructs of the HBM) at bivariate analyses (Chi-square) were subsequently added to the multivariable binary logistic regression models one at a time while adjusting for the confounding effects of other independent variables (sociodemographic variables & the constructs of the HBM). The regression results were presented using adjusted odds ratios (AOR) with their respective 95% confidence intervals.

### Ethical issues

The University of Health and Allied Sciences Research Ethics Committee (UHAS-REC) reviewed and approved the study (**UHAS-REC B.10 [163] 21–22).** In addition, the study sought permission from GNMHD, and participants were only included in the study after full written consent. All participants were aged 18 years and above, so there was no minor in the study to warrant their written assent and parental consent. All completed hardcopy questionnaires and signed informed consent forms were kept confidential in a lockable cabinet only to be accessed by the authors. Softcopy data were kept on a computer with restricted access. Only the authors had access to the data. Participants' names were not written on the questionnaires and all linking personal details of participants were re-coded to ensure anonymity.

## Results

### Sociodemographic characteristics of respondents

A total of 388 respondents participated in this study with a 100% response rate. This response rate was achieved possibly because the purpose of the study was duly explained to the respondents and the principal investigator and the research assistant were present to clarify issues and provide answers to questions posed by the respondents during data collection. The majority (53.4%) were females and the majority (43.0%) were less than 30 years of age with a median age of 31 years and an interquartile range of 25–41 years. A majority (38.1%) had tertiary education and most (50.5%) were married. Close to half (48.2%) had a household size of less than 4 and a majority (70.9%) were Christians. Most (93.8%) resided in urban settings; a few (17.3%) had a history of chronic disease and cardiovascular disease (8.3%) was reported more among persons with a history of chronic conditions. The majority (57.0%) reported none of their household members had any of those chronic conditions and just a few (20.9%) reported having a family member who had been confirmed positive for COVID-19 (**Table 2).**

### COVID-19 vaccine uptake

The majority of respondents (72.2%) had received the COVID-19 vaccine prior to the study.

### Health Belief Model constructs influencing COVID-19 vaccination uptake

The majority of the respondents (71.9%) and (78.4%) believed they could contract COVID-19 and had increased chances of contracting the disease if unvaccinated, respectively. Most (82.0%) believed if unvaccinated, they have increased chances of infecting their family members and most (73.2%) had the fear of being infected with COVID-19. The majority (68.6%) believed they can develop complications if infected with COVID-19 and most (58.0%) believed they are likely to die if infected with COVID-19 (perceived threat) (**Table 3).**

From Table 4, the majority (67.3%) believed vaccination would prevent them from contracting COVID-19. The majority (79.4%) believed if vaccinated they would be protected from

**Table 2. Sociodemographic characteristics of respondents (n = 388).**

| Variables | Frequency (n) | Percentage (%) |
|---|---|---|
| **Sex** | | |
| Female | 207 | 53.3 |
| Male | 181 | 46.7 |
| **Age (years)** | | |
| < 30 | 168 | 43.3 |
| 30–49 | 167 | 43.0 |
| 50 + | 53 | 13.7 |
| **Median age (Interquartile range)** | 31 (25–41) | |
| **Educational level** | | |
| No formal education | 45 | 11.6 |
| Basic education | 90 | 23.2 |
| SHS | 105 | 27.1 |
| Tertiary | 148 | 38.1 |
| **Marital Status** | | |
| Married | 196 | 50.5 |
| Single | 192 | 49.5 |
| **Household Size** | | |
| <4 | 187 | 48.1 |
| 4–6 | 169 | 43.6 |
| 7+ | 32 | 8.3 |
| **Religion** | | |
| Christian | 275 | 70.9 |
| Muslim | 94 | 24.2 |
| Traditionalist | 19 | 4.9 |
| **Place of residence** | | |
| Rural | 24 | 6.2 |
| Urban | 364 | 93.8 |
| **Any chronic disease** | | |
| No | 321 | 82.7 |
| Yes | 67 | 17.3 |
| **History of chronic diseases*** | | |
| Cancer | 18 | 4.6 |
| Cardiovascular disease | 32 | 8.3 |
| Diabetes | 25 | 6.4 |
| Hepatic disease | 6 | 1.6 |
| Others | 19 | 4.9 |
| **Household member(s) with any of the above disease** | | |
| No | 221 | 57.0 |
| Yes | 167 | 43.0 |
| **Family member confirmed positive for COVID-19** | | |
| No | 307 | 79.1 |
| Yes | 81 | 20.9 |

*Multiple responses

**Table 3. Perceived threat by the respondents.**

| Variables | Frequency (n) | Percentage (%) |
|---|---|---|
| **Perceived susceptibility construct** | | |
| **I can contract COVID-19** | | |
| Agree | 279 | 71.9 |
| Disagree | 109 | 28.1 |
| **Increased chances of COVID-19 if unvaccinated** | | |
| Agree | 304 | 78.4 |
| Disagree | 84 | 21.6 |
| **Increased chances of infecting family members if unvaccinated** | | |
| Agree | 318 | 82.0 |
| Disagree | 70 | 18.0 |
| **Fear of getting infected with COVID-19** | | |
| Agree | 284 | 73.2 |
| Disagree | 104 | 26.8 |
| **Perceived Severity Construct** | | |
| **Low chances of recovering from COVID-19 if infected** | | |
| Agree | 182 | 46.9 |
| Disagree | 206 | 53.1 |
| **Develop complications if infected with COVID-19** | | |
| Agree | 266 | 68.6 |
| Disagree | 122 | 31.4 |
| **Likely to die if infected with COVID-19** | | |
| Agree | 225 | 58.0 |
| Disagree | 163 | 42.0 |

developing complications. Most (54.9%) perceived the side effects of the vaccine as a barrier to them. The majority (58.8%) believed COVID-19 vaccines cause complications and less than half (43.8%) believed the vaccines are ineffective (Behavioural evaluation).

Results from Table 5 show that the majority (84.3%) believed they had enough information on COVID-19 and its vaccines. Most (76.0%) were willing to uptake COVID-19 vaccination when given adequate information on the disease and its vaccines. The majority (70.9%) and (68.3%) were willing to uptake COVID-19 vaccination if opinion leaders, and family and friends express the benefits of vaccines, respectively, and most (66.2%) would vaccinate if vaccination is mandatory (cues to action). Most (72.7%) were willing to vaccinate if vaccines were made available. The majority (72.9%) were confident to vaccinate regardless of side effects and most (77.8%) were confident to vaccinate despite reluctance of opinion leaders, family and friends to vaccinate (self-efficacy).

## Theoretical influence (HBM) of behavioural response to COVID-19 vaccine uptake

The overall behavioural responses to COVID-19 vaccine uptake showed that the majority (72.2%) believed that being vaccinated would reduce their chances of contracting COVID-19 and most (61.3%) believed that vaccination can reduce complications from COVID-19. The majority (63.7%) perceived that COVID-19 vaccination would be beneficial to them and 54.6% perceived certain barriers to COVID-19 vaccine uptake. The majority (70.4%) believed that cues would trigger them and 62.6% had the confidence to uptake the COVID-19 vaccine (Fig 2).

Table 4. Behavioural evaluation by the respondents.

| Perceived benefit construct | Frequency (n) | Percentage (%) |
|---|---|---|
| **Prevention against COVID-19 if vaccinated** | | |
| Agree | 261 | 67.3 |
| Disagree | 127 | 32.7 |
| **Protection from developing complications if vaccinated** | | |
| Agree | 308 | 79.4 |
| Disagree | 80 | 20.6 |
| **Perceived barrier construct** | | |
| **Far distance to vaccination site as a barrier to COVID-19 vaccination** | | |
| Agree | 172 | 44.3 |
| Disagree | 216 | 55.7 |
| **Side effects of vaccines** | | |
| Agree | 213 | 54.9 |
| Disagree | 175 | 45.1 |
| **Vaccines are ineffective** | | |
| Agree | 170 | 43.8 |
| Disagree | 218 | 65.2 |
| **Busy schedules** | | |
| Agree | 126 | 32.5 |
| Disagree | 262 | 67.5 |
| **Mistrust of vaccines** | | |
| Agree | 187 | 48.2 |
| Disagree | 201 | 51.8 |
| **COVID-19 vaccines cause complications** | | |
| Agree | 228 | 58.8 |
| Disagree | 160 | 41.2 |

## Sociodemographic predictors of COVID-19 vaccine uptake

Table 6 presents the results of the bivariate and multivariate analyses of socio-demographic predictors of COVID-19 vaccine uptake. The results showed that the odds of COVID-19 vaccination uptake were higher among men than women [AOR = 2.02, 95% Cl: 1.13–3.20]. Single respondents had 2 times the odds of receiving COVID-19 vaccines compared to those who were married [AOR = 1.90, 95% Cl: 1.07–3.36]. Respondents who were Muslims were less likely to receive COVID-19 vaccination than the Christians [AOR = 0.33, 95%Cl: 0.18–0.60] (**Table 6**).

## Health Belief Model constructs predicting COVID-19 vaccine uptake

After controlling for covariates in the multivariate analysis, perceived susceptibility, perceived barriers, cues to actions and self-efficacy were the constructs predicting COVID-19 vaccine uptake. Respondents who believed they were susceptible to COVID-19 had twice the odds of receiving COVID-19 vaccines [AOR = 2.43, 95% Cl: 1.36–4.35]. Persons with perceived barriers to COVID-19 vaccine uptake were 46% less likely to uptake the COVID-19 vaccine [AOR = 0.54, 95%Cl: 0.31–0.95]]. Respondents with stronger cues for COVID-19 vaccine uptake had 2 times the odds of receiving the COVID-19 vaccine [AOR = 2.23, 95% Cl: 1.19–4.21] and respondents who had self-confidence were three times more likely to receive COVID-19 vaccine [AOR = 3.23 95% Cl: 1.82–5.71] (**Table 7**).

**Table 5. Cues to action and self-efficacy by the respondents.**

| Cues to action construct | Frequency (n) | Percentage (%) |
|---|---|---|
| **Enough information on COVID-19 and its vaccines** | | |
| Agree | 327 | 84.3 |
| Disagree | 61 | 15.7 |
| **Acceptance of COVID-19 vaccination when given adequate information on COVID-19 and its vaccines** | | |
| Agree | 295 | 76.0 |
| Disagree | 93 | 24.0 |
| **Acceptance of COVID-19 vaccination if opinion leaders express support for the benefits of vaccines** | | |
| Agree | 275 | 70.9 |
| Disagree | 113 | 29.1 |
| **Acceptance of vaccination if family and friends express support for the benefits of vaccines** | | |
| Agree | 265 | 68.3 |
| Disagree | 123 | 31.7 |
| **Acceptance of vaccination if mandatory** | | |
| Agree | 257 | 66.2 |
| Disagree | 131 | 33.8 |
| **Self-efficacy construct** | | |
| **Willingness to vaccinate if the COVID-19 vaccine is made available** | | |
| Agree | 282 | 72.7 |
| Disagree | 106 | 27.3 |
| **Confidence to vaccinate despite the side effects of vaccines** | | |
| Agree | 283 | 72.9 |
| Disagree | 105 | 27.1 |
| **Confidence to vaccinate despite the reluctance of opinion leaders, family and friends to vaccinate** | | |
| Agree | 302 | 77.8 |
| Disagree | 86 | 22.2 |

## Discussion

The current study examined the predictors of COVID-19 vaccine uptake among adults in the Ga North Municipality, using the HBM. In the current study, vaccination uptake was approximately 72.2%, which is higher than the findings of Echoru et al. (2021) [30], which reported 54% of COVID-19 vaccination uptake in Uganda and 39% prevalence of COVID-19 vaccination uptake in Ghana [15]. The differences between the period for the conduct of the present study relative to those studies could have accounted for significant variations in the uptake of the COVID-19 vaccine. Additionally, the adults in the Ga North Municipality could have received more information from mass media platforms, which could have helped to avert most of the negative misconceptions they might have concerning the COVID-19 vaccines, thereby increasing their uptake.

### Sociodemographic predictors of COVID-19 vaccine uptake

The sex of the respondent was found to be significantly associated with COVID-19 vaccine uptake in the current study. Males have higher odds of COVID-19 vaccine uptake compared

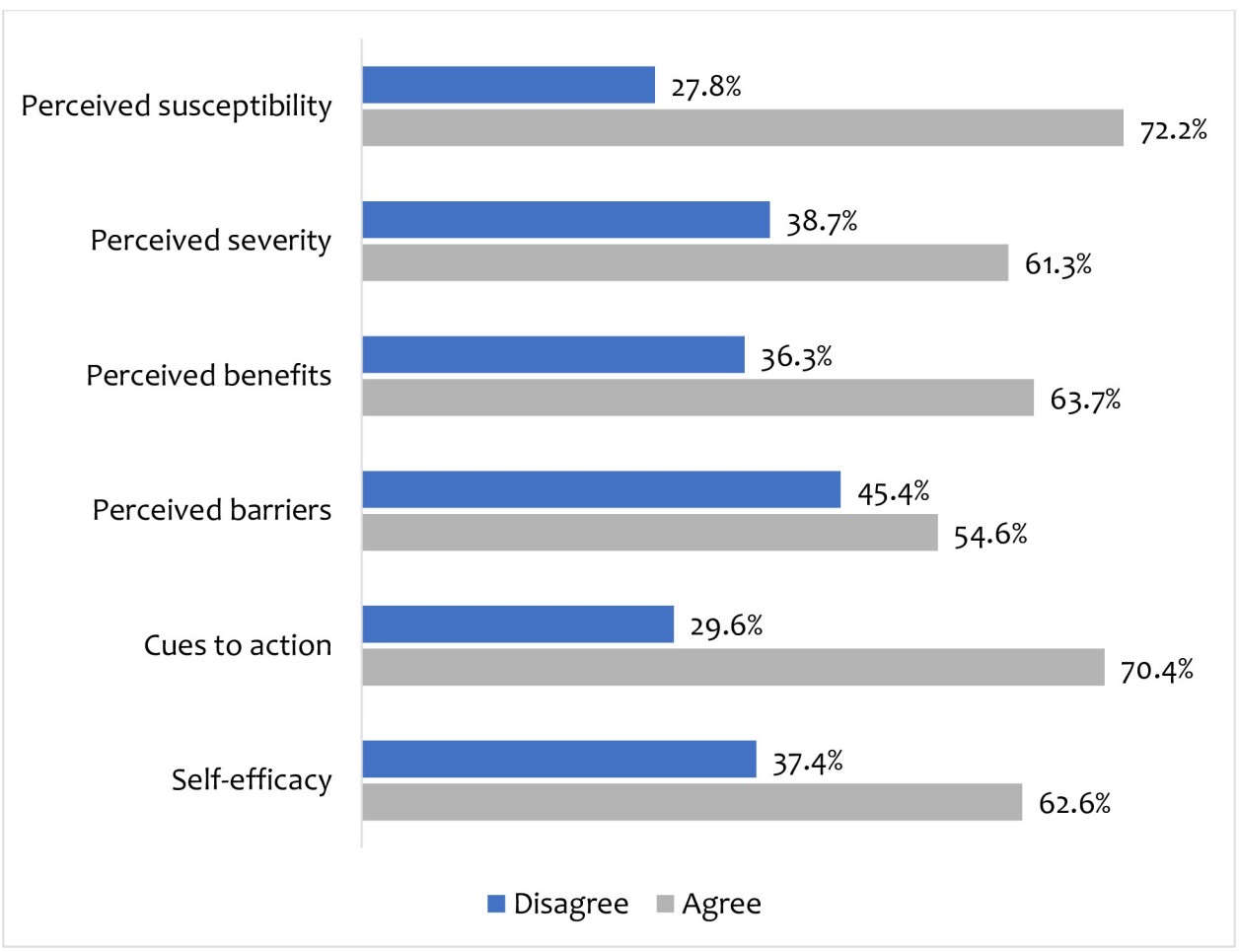

**Fig 2. COVID-19 vaccine uptake-related health beliefs.**

to females, which is in tandem with previous findings [16, 27, 31, 32]. Higher odds of vaccine uptake among males have been associated with a higher risk of severity and mortality in males than females and fear of vaccine safety by the female gender [30, 33]. Mechanisms that serve as protective factors for females have been ascribed to hormonal differences in both sexes [34–36]. The current finding, thus calls for interventions to increase COVID-19 vaccine uptake targeting the female population.

Religious influence on COVID-19 vaccine uptake was reported in the current study with Muslims being less likely to receive the vaccination. This corroborates with Razai et al. (2021) [37] who reported low vaccine uptake among Muslims due to their belief that vaccines are 'haram'. Contrarily, Echoru et al. (2021) reported increased odds of COVID-19 vaccine uptake among Muslims than Christians [30]. These differential findings may be associated with the quantum of these religions' dominance in the settings the different studies were conducted.

The current study also revealed that single respondents had 2 times the odds of receiving COVID-19 vaccines compared to those who were married. Similar findings have been reported by Achangwa et al. (2021) [38]. Therefore, there is a need for health promotion intervention to increase COVID-19 vaccine uptake among the Muslim and married population.

**Table 6. Sociodemographic predictors of COVID-19 vaccine uptake.**

| Variables | COVID-19 Vaccination uptake | | Chi-square (χ2) | p-value | AOR (95% Cl) p-value |
|---|---|---|---|---|---|
| | No = [108] n (%) | Yes [280] n (%) | | | |
| **Sex** | | | 4.54 | 0.033 | |
| Female | 67(62.0) | 140(50.0) | | | Ref |
| Male | 41(38.0) | 140(50.0) | | | 2.02[1.13–3.20], 0.015 |
| **Age [Years]** | | | 7.95 | 0.019 | |
| <30 | 37(34.3) | 131(46.8) | | | |
| 30–49 | 49(45.4) | 118(42.1) | | | |
| 50 + | 22(20.4) | 31(11.0) | | | |
| **Educational level** | | | 10.65 | 0.014 | |
| No formal education | 19(17.6) | 26(9.26) | | | |
| Basic school | 29(26.9) | 61(21.8) | | | |
| SHS | 31(28.7) | 74(26.4) | | | |
| Tertiary | 29(26.9) | 119(42.5) | | | |
| **Marital status** | | | 5.60 | 0.018 | |
| Married | 65(60.2) | 131(46.8) | | | Ref |
| Single | 43(39.9) | 149(53.2) | | | 1.90[1.07–3.36], 0.028 |
| **Religion** | | | | | |
| Christian | 65(60.2) | 210(75.0) | 8.30 | 0.016 | Ref |
| Muslim | 36(33.3) | 58(20.7) | | | 0.33[0.18–0.60], <0.001 |
| Traditionalist | 7(6.48) | 12(4.3) | | | 0.32[0.10–1.01], 0.052 |
| **Place of residence** | | | | | |
| Rural | 12(11.1) | 12(4.3) | 6.26 | 0.012 | |
| Urban | 96(88.9) | 268(95.7) | | | |
| **Household member with any chronic disease** | | | 12.55 | <0.001 | |
| No | 77(71.3) | 144(51.4) | | | |
| Yes | 31(28.7) | 136(48.6) | | | |
| **Family member confirmed positive for COVID-19** | | | 18.77 | <0.001 | |
| No | 101(93.5) | 206(73.6) | | | |
| Yes | 7(6.5) | 74(26.4) | | | |

## Health Belief Model constructs predicting COVID-19 vaccination uptake

After controlling for covariates in the multivariate analysis, perceived susceptibility, perceived barriers, cues to actions and self-efficacy were the constructs significantly associated with COVID-19 vaccine uptake.

In the current study, respondents who believed they were susceptible to COVID-19 infection had twice the odds of receiving COVID-19 vaccines. This is consistent with the findings of Wong et al. (2020) [19]. This finding may be attributed to individuals' perception of COVID-19 as a personally relevant problem with high vulnerability, thus will adopt protective measures including uptake of COVID-19 vaccination. It is, therefore, prudent to increase SARS-CoV-2 susceptibility communication among the population to increase COVID-19 vaccine uptake [17, 39]. However, this finding is contrary to that of Shmueli (2021) [16] who reported no association between risk perception and COVID-19 vaccine uptake. These observed differential findings may be attributed to differences in demographic characteristics of respondents and study settings vis-à-vis reported high susceptibility to COVID-19 among the aged compared to young persons [10].

The current study also reported a significant association between perceived barriers to COVID-19 vaccination uptake and COVID-19 vaccination uptake in that those who perceived

**Table 7. Health Belief Model constructs predicting COVID-19 vaccine uptake.**

| Variables | COVID-19 Vaccination uptake | | Chi-square (χ2) | p-value | AOR (95% CI) p-value |
|---|---|---|---|---|---|
| | No = [108] n (%) | Yes [280] n (%) | | | |
| **Perceived susceptibility** | | | 46.35 | <0.001 | |
| No | 57(52.8) | 51(18.2) | | | Ref |
| Yes | 51(47.2) | 229(81.8) | | | 2.43[1.36–4.35], 0.003 |
| **Perceived severity** | | | 14.28 | <0.001 | |
| No | 58(53.7) | 92(32.9) | | | |
| Yes | 50(46.3) | 188(67.1) | | | |
| **Perceived benefits** | | | 31.29 | <0.001 | |
| No | 63(58.3) | 78(27.9) | | | |
| Yes | 45(41.7) | 202(72.1) | | | |
| **Perceived barriers** | | | 8.74 | 0.003 | |
| No | 36(33.3) | 140(50.0) | | | Ref |
| Yes | 72(66.7) | 140(50.0) | | | 0.54[0.31–0.95], 0.031 |
| **Cues to action** | | | 44.8 | <0.001 | |
| No | 59(54.6) | 56(20.0) | | | Ref |
| Yes | 49(45.4) | 224(80.0) | | | 2.23[1.19–4.21], 0.012 |
| **Self-efficacy** | | | 65.77 | <0.001 | |
| No | 75(69.4) | 70(25.0) | | | Ref |
| Yes | 33(30.6) | 210(75.0) | | | 3.23[1.82–5.71], <0.001 |

some barriers were less likely to uptake the vaccine. This finding corroborates those of similar studies [24, 32, 40], which have reported vaccine-related barriers as increasing COVID-19 vaccine hesitancy. This finding calls for health promotion interventions to minimise these barriers through correcting misperceptions about the vaccine, providing incentives and providing skills for COVID-19 vaccine uptake.

Cues to action are defined as events and/or experiences that may heighten or increase awareness to trigger acceptance of the COVID-19 vaccine. This may include access to adequate information on COVID-19 and its vaccines and societal acceptance of the vaccine [26, 39]. In the current study, it was reported that individuals with stronger cues had 2 times the odds of receiving COVID-19 vaccination. This is consistent with the findings of Walker et al. (2021) [18] and Wong et al. (2020) [19]. The provision of adequate information on COVID-19 and its vaccines will help neutralise misconceptions such as claims that COVID-19 vaccines contain microchips, cause infertility or death, alter human DNA, and claim that COVID-19 is a bio-weapon. This could in turn improve confidence in the vaccine and subsequent uptake [41, 42]. Hence the need to leverage all platforms including HCWs to communicate COVID-19-focused information to the population effectively [15].

Self-efficacy refers to the confidence an individual has in his/her ability to adhere to protected measures against COVID-19. High self-efficacy can facilitate uptake of COVID-19 vaccination [17, 39]. This study has revealed that persons with self-confidence were three times more likely to uptake COVID-19 vaccination compared to their counterparts with low self-efficacy. Similarly, Stout, Christy, Winger, Vadaparampil, and Mosher (2020) [43] reported a positive association between COVID-19 vaccination and the attainment of self-efficacy. The current finding was also consistent with Walker et al. (2021) who reported a significant association between low self-confidence and vaccine hesitancy [18]. The current finding calls for interventions to increase self-efficacy for COVID-19 vaccine uptake. Contrary to these

findings, Bock et al. (2017) [44] have reported a correlation between low self-esteem and acceptance of vaccination. The reason for this can be attributed to low self-esteem persons' proven inability to use cognitive measures to reduce potential risk compared to high self-esteem persons who can prevent infection without vaccination [45, 46].

## Limitations of the study

The current findings are not without limitations. First, the design used cannot predict past and future changes in studied variables since it was a one-time study but not a longitudinal study. Second, proportionate selection of socio-demographic variables such as age, gender, and religious affiliation was not considered in the current study. Despite these limitations, the current study included a proportionate selection of respondents from all sub-municipalities, thus ensuring representativeness and making the present findings generalisable. To the best of our knowledge, this is the first study to determine the predictors of COVID-19 vaccine uptake using the HBM in the Municipality.

## Conclusion

Although vaccination uptake is high, about 27.8% of the study population remains unvaccinated against SARS-CoV-2. Sex, religion, and marital status were found to influence COVID-19 vaccine uptake. In addition, individuals' perceived susceptibility to SARS-CoV-2; perceived barriers to COVID-19 vaccine uptake; cues to action; and self-efficacy for COVID-19 vaccine were the significant predictors of COVID-19 vaccination uptake. Thus, there is a need to develop and implement policies aimed at addressing these predictors and improving COVID-19 vaccine uptake.

## Supporting information

**S1 Data.**
(XLSX)

## Author Contributions

**Conceptualization:** Imoro Nasiratu, Elvis Enowbeyang Tarkang.

**Data curation:** Imoro Nasiratu, Elvis Enowbeyang Tarkang.

**Formal analysis:** Imoro Nasiratu, Elvis Enowbeyang Tarkang.

**Investigation:** Imoro Nasiratu, Elvis Enowbeyang Tarkang.

**Methodology:** Imoro Nasiratu, Elvis Enowbeyang Tarkang.

**Supervision:** Elvis Enowbeyang Tarkang.

**Validation:** Lilian Belole Pencille, Nelisiwe Khuzwayo, Richard Gyan Aboagye, Elvis Enowbeyang Tarkang.

**Writing – original draft:** Imoro Nasiratu, Nelisiwe Khuzwayo, Richard Gyan Aboagye, Elvis Enowbeyang Tarkang.

**Writing – review & editing:** Imoro Nasiratu, Lilian Belole Pencille, Nelisiwe Khuzwayo, Richard Gyan Aboagye, Elvis Enowbeyang Tarkang.

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
