## [Decision Letter · Decision Letter 0]

24 Aug 2023

PONE-D-23-10818Predictors of COVID-19 vaccine uptake among persons aged 18 years and above in Ga North Municipality, Ghana using the Health Belief Model: a community-based cross-sectional studyPLOS ONE

Dear Dr. Elvis Enowbeyang Tarkang,

Thank you for submitting your manuscript to PLOS ONE. After careful consideration, we feel that it has merit but does not fully meet PLOS ONE’s publication criteria as it currently stands. Therefore, we invite you to submit a revised version of the manuscript that addresses the points raised during the review process.

Please submit your revised manuscript 14 September, 2023. If you will need more time than this to complete your revisions, please reply to this message or contact the journal office at plosone@plos.org. Please include the following items when submitting your revised manuscript:A rebuttal letter that responds to each point raised by the academic editor and reviewer(s). You should upload this letter as a separate file labeled 'Response to Reviewers'.A marked-up copy of your manuscript that highlights changes made to the original version. You should upload this as a separate file labeled 'Revised Manuscript with Track Changes'.An unmarked version of your revised paper without tracked changes. You should upload this as a separate file labeled 'Manuscript'.If applicable, we recommend that you deposit your laboratory protocols in protocols.io to enhance the reproducibility of your results. Protocols.io assigns your protocol its own identifier (DOI) so that it can be cited independently in the future. For instructions see: https://journals.plos.org/plosone/s/submission-guidelines#loc-laboratory-protocols. Additionally, PLOS ONE offers an option for publishing peer-reviewed Lab Protocol articles, which describe protocols hosted on protocols.io. Read more information on sharing protocols at https://plos.org/protocols?utm_medium=editorial-email&utm_source=authorletters&utm_campaign=protocols.

We look forward to receiving your revised manuscript.

Kind regards,

Dr. Oluwatosin Oluwaseun Olu-Abiodun

Academic Editor

PLOS ONE

Journal Requirements:

2. PLOS requires an ORCID iD for the corresponding author in Editorial Manager on papers submitted after December 6th, 2016. Please ensure that you have an ORCID iD and that it is validated in Editorial Manager. To do this, go to ‘Update my Information’ (in the upper left-hand corner of the main menu), and click on the Fetch/Validate link next to the ORCID field. This will take you to the ORCID site and allow you to create a new iD or authenticate a pre-existing iD in Editorial Manager. Please see the following video for instructions on linking an ORCID iD to your Editorial Manager account: https://www.youtube.com/watch?v=_xcclfuvtxQ.

Reviewers' comments:

Reviewer's Responses to Questions

**Comments to the Author**

1. Is the manuscript technically sound, and do the data support the conclusions?

Reviewer #1: Yes

Reviewer #2: Yes

Reviewer #3: Yes

2. Has the statistical analysis been performed appropriately and rigorously? 

Reviewer #1: Yes

Reviewer #2: Yes

Reviewer #3: Yes

3. Have the authors made all data underlying the findings in their manuscript fully available?

Reviewer #1: Yes

Reviewer #2: No

Reviewer #3: Yes

4. Is the manuscript presented in an intelligible fashion and written in standard English?

Reviewer #1: Yes

Reviewer #2: Yes

Reviewer #3: Yes

5. Review Comments to the Author

Reviewer #1: The manuscript is well written but there are some minor corrections. I suggest that the authors rewrite the background to reflect the variables that were studied. The authors wrote about vaccine hesitancy at the third paragraph and had little write-up as it relates to factors influencing vaccine uptake. In addition the writers wrote extensively on health care workers when they are not the main population. There was a spelling mistake in the last sentence before the conclusion, the letter 'e' is missing from the predictors. I suggest that you change the statement the population of the non-vaccinated people is high because the study did not cover the number of people who are vaccinated, the study only covered uptake. Furthermore, the percentage of the uptake is high. Therefore I suggest that it is changed.

Reviewer #2: All relevant data should be included in the study. Avoid duplication of data. The non-vaccinated group should be statistically identified. Studies can also be conducted on the other groups. Health promotion interventions should can be extended further to eradicate barriers in taking covid-19 vaccines and other related vaccines.

Reviewer #3: I have had the privilege of reviewing your manuscript. The study was well-conceptualized, conducted and written. The use of the health belief model was also appropriate. I have a few critiques for you to consider to improve the quality of the write-up to meet international readability standards.

The manuscript will benefit from a general review to mitigate grammatical and syntax errors. The study background seems rather lengthy. Could it be more precise? Focusing the write-up entirely on the subject matter (COVID-19 vaccine uptake) may help. Since this is a cross-sectional study, why did you require the participants to intend to live in the municipality to be eligible? The description of the study population seems more like the inclusion criteria, which is then repeated immediately after. The formula you use for sample size determination has been attributed to another set of authors. This is the first time I am seeing it attributed to Degu and Tessema formula. Kindly verify this and adjust if necessary. What motivated the choice of 5% as the non-response rate? Why were the sample sizes disproportionately allocated among selected communities? Did you mean to say proportionately allocated? What is your outlook on the coefficient for internal consistency (alpha Cronbach?) less than 0.7? How does this affect the interpretation of the perceived severity and benefit domains? Could these values have been improved by dropping some of the items in the domains? You had a 100% response rate with no missing data. Are there specific lessons other researchers (especially in the sub-Saharan African context) could learn to help achieve these? It is critical to underscore this detail in the study. You used relevant literature quite well but could also consider other regional findings from studies like https://doi.org/10.1371/journal.pone.0267691

Thank you for the good study!

6. PLOS authors have the option to publish the peer review history of their article (what does this mean?). If published, this will include your full peer review and any attached files.

Reviewer #1: No

Reviewer #2: **Yes: **SODIMU JEMINAT OMOTADE (RN, Ph.D.)

Reviewer #3: **Yes: **Olumide Abiodun

---

## [Author Response · Author response to Decision Letter 0]

18 Sep 2023

Reviewer 1

The manuscript is well written but there are some minor corrections. 

Comment

I suggest that the authors rewrite the background to reflect the variables that were studied.

Response

Thank you for this comment. The background has been rewritten to reflect the variables that were studied. 

Comment

The authors wrote about vaccine hesitancy at the third paragraph and had little write-up as it relates to factors influencing vaccine uptake.

Response

Thank you for this comment. This has been addressed. 

Comment

In addition, the writers wrote extensively on health care workers when they are not the main population.

Response

This comment has been addressed.

Comment

There was a spelling mistake in the last sentence before the conclusion, the letter 'e' is missing from the predictors. I suggest that you change the statement the population of the non-vaccinated people is high because the study did not cover the number of people who are vaccinated, the study only covered uptake. Furthermore, the percentage of the uptake is high. Therefore, I suggest that it is changed.

Response

Thank you very much for these important comments. The issues raised have been duly addressed. 

Reviewer 2 

Comment

All relevant data should be included in the study. Avoid duplication of data.

Response

Thank you for this comment. This has been addressed; all relevant data have been included as an appendix 

Comment

The non-vaccinated group should be statistically identified. Studies can also be conducted on the other groups. 

Response

Thank you for this comment. The non-vaccinated group made up 27.8% of the study population statistically. We agree with your comment that further studies can also be conducted on them and other groups subsequently.

Comment

Health promotion interventions should can be extended further to eradicate barriers in taking covid-19 vaccines and other related vaccines.

Response

Thank you very much for this comment. Our study determined the predictors of COVID-19 vaccine uptake based on the constructs of the Health Belief Model, of which barriers are part. From our findings, barriers were among the significant predictors of vaccine uptake and recommendations were made in this regard. Studies can be conducted regarding the uptake of other vaccines if there is a need for that. 

Reviewer 3

General comments

I have had the privilege of reviewing your manuscript. The study was well-conceptualized, conducted and written. The use of the health belief model was also appropriate. I have a few critiques for you to consider to improve the quality of the write-up to meet international readability standards.

Thank you for the good study!

Response

The authors are appreciative of your kind words and commendation. Thank you for the review. The authors have duly acknowledged and addressed your comments. 

Comments

The manuscript will benefit from a general review to mitigate grammatical and syntax errors.

Response

Grammatical errors have been corrected.

Comment

The study background seems rather lengthy. Could it be more precise? Focusing the write-up entirely on the subject matter (COVID-19 vaccine uptake) may help. 

Response 

Thank you very much for this comment. We have addressed the issue. 

Comments 

Since this is a cross-sectional study, why did you require the participants to intend to live in the municipality to be eligible? The description of the study population seems more like the inclusion criteria, which is then repeated immediately after.

Response

The comments have been noted and duly addressed. 

Comment

The formula you use for sample size determination has been attributed to another set of authors. This is the first time I am seeing it attributed to Degu and Tessema formula. Kindly verify this and adjust if necessary. 

Response

Thank you so much for this very important comment, which has now been addressed. 

Comments

What motivated the choice of 5% as the non-response rate?

Response

This was chosen because of the sensitive nature of the study, which could lead to an increased dropout rate of the participants. 5% has been used in a previous study that I have now cited.

Comment

Why were the sample sizes disproportionately allocated among selected communities? Did you mean to say proportionately allocated?

Response

This has been addressed. 

Comment

What is your outlook on the coefficient for internal consistency (alpha Cronbach?) less than 0.7? How does this affect the interpretation of the perceived severity and benefit domains? Could these values have been improved by dropping some of the items in the domains?

Response

The overall alpha coefficient for the entire HBM was 0.8554 which is excellent. The coefficients were calculated using the best combination of items under each construct of the HBM, excluding the response options. Cronbach's alpha does not provide a very good estimate when the items making up the measurement scale are small in number. The more items to measure a construct the more accurate the coefficient. Therefore, the relatively low alpha estimated for perceived severity (0.6759), may be due to the small number of items measuring this construct (3 items). 

Comment

You had a 100% response rate with no missing data. Are there specific lessons other researchers (especially in the sub-Saharan African context) could learn to help achieve these? It is critical to underscore this detail in the study.

Response

This response rate was achieved possibly because the purpose of the study was duly explained to the respondents and the principal investigator and the research assistant were present to clarify issues and provide answers to questions posed by the respondents during data collection. 

Comment

You used relevant literature quite well but could also consider other regional findings from studies like https://doi.org/10.1371/journal.pone.0267691.

Response

This point has been noted and the recommended reference and others have been included in a subsequent paper that we are drafting. The current study focused on vaccine uptake and its predictors based on the Health Belief Model. The recommended paper above was a rapid review of papers published on COVID-19 vaccine acceptance and associated factors in Nigeria. There is a difference between vaccine acceptance and vaccine uptake. The paper would fit well in the manuscript we are drafting.

---

## [Editor Report · Decision Letter 1]

11 Oct 2023

Predictors of COVID-19 vaccine uptake among persons aged 18 years and above in Ga North Municipality, Ghana using the Health Belief Model: a community-based cross-sectional study

PONE-D-23-10818R1

Dear Dr. Elvis Enowbeyang Tarkang,

We’re pleased to inform you that your manuscript has been judged scientifically suitable for publication and will be formally accepted for publication once it meets all outstanding technical requirements.

Kind regards,

Dr. Oluwatosin Olu-Abiodun

Academic Editor

PLOS ONE
---

## [Editor Report · Acceptance letter]

30 Oct 2023

PONE-D-23-10818R1 

Predictors of COVID-19 vaccine uptake among persons aged 18 years and above in Ga North Municipality, Ghana using the Health Belief Model: a community-based cross-sectional study 

Dear Dr. Tarkang:

I'm pleased to inform you that your manuscript has been deemed suitable for publication in PLOS ONE. Congratulations! Your manuscript is now with our production department. 

Kind regards, 

on behalf of

Dr. Oluwatosin Oluwaseun Olu-Abiodun 

Academic Editor

PLOS ONE